# Polymorphisms Within the *IQGAP2* and *CRTAC1* Genes of Gannan Yaks and Their Association with Milk Quality Characteristics

**DOI:** 10.3390/foods13233720

**Published:** 2024-11-21

**Authors:** Juanxiang Zhang, Guowu Yang, Xita Zha, Xiaoming Ma, Yongfu La, Xiaoyun Wu, Xian Guo, Min Chu, Pengjia Bao, Ping Yan, Chunnian Liang

**Affiliations:** 1Key Laboratory of Yak Breeding Engineering of Gansu Province, Lanzhou Institute of Husbandry and Pharmaceutical Sciences, Chinese Academy of Agricultural Sciences, Lanzhou 730050, China; 15103990593@163.com (J.Z.); guowu202302@163.com (G.Y.); maxiaoming@caas.cn (X.M.); layongfu@caas.cn (Y.L.); wuxiaoyun@caas.cn (X.W.); guoxian@caas.cn (X.G.); chumin@caas.cn (M.C.); baopengjia@caas.cn (P.B.); pingyanlz@163.com (P.Y.); 2Key Laboratory of Animal Genetics and Breeding on Tibetan Plateau, Ministry of Agriculture and Rural Affairs, Lanzhou 730050, China; 3Qinghai Qilian County Animal Husbandry and Veterinary Workstation, Qilian 810400, China; zhaxita@163.com

**Keywords:** *IQGAP*, *CRTAC1*, milk quality traits, SNPs, Gannan yak

## Abstract

The IQ motif containing GTPase activating protein 2 (*IQGAP2*) gene functions as a tumor suppressor, reducing the malignant properties of breast cancer cells. The circulating cartilage acidic protein 1 (*CRTAC1*) gene, present in the whey protein fraction of dairy cows throughout lactation, is significantly correlated with fatty acids in milk. In this study, we investigated the correlation between single nucleotide polymorphisms (SNPs) in the *IQGAP2* and *CRTAC1* genes and milk quality traits in Gannan yaks, aiming to identify potential molecular marker loci for enhancing milk quality. Using the Illumina Yak cGPS 7K liquid chip, we genotyped 162 yaks and identified five SNPs in the *IQGAP2* (g.232,769C>G, g.232,922G>C) and *CRTAC1* (g.4,203T>C, g.5,348T>G, g.122,451T>C) genes. Genetic polymorphism analysis revealed that these five SNPs were moderately polymorphic and in Hardy–Weinberg equilibrium. An association analysis results showed that, at the g.232,769C>G locus of the *IQGAP2* gene, the heterozygous CG genotype had significantly higher lactose content than the CC and GG homozygous genotypes (*p <* 0.05). Similarly, at the g.232,922G>C locus, the heterozygous GC and mutant CC genotypes significantly increased the contents of milk fat, lactose, and total solids (TS) (*p* < 0.05). In the *CRTAC1* gene (g.4,203T>C, g.5,348T>G, g.122,451T>C), the mutant CC genotype significantly increased milk fat content, while the heterozygous TG genotype significantly increased lactose content (*p* < 0.05). In summary, mutations at the loci of g.232,769C>G, g.232,922G>C, g.4,203T>C, g.5,348T>G, and g.122,451T>C significantly elevated the lactose, milk fat, and TS content in Gannan yak milk, providing potential molecular marker candidates for improving Gannan yak milk quality.

## 1. Introduction

Yaks inhabit the Qinghai–Tibet Plateau and adjacent high-altitude regions, exhibiting remarkable adaptability to harsh environments defined by low temperatures, elevated altitudes, and intense ultraviolet radiation. These animals play a crucial role as a vital source of meat, milk, and other daily necessities for local herdsmen [1]. Renowned as the “boats of the plateau” and the “all-purpose livestock”, they constitute the backbone of the local livestock industry and hold a pivotal position in economic development [2]. As a representative of plateau-characteristic livestock products, yak milk has a high content of fat (5.5–7.5%), protein (4.0–5.9%), and lactose (4.0–5.9%) during the peak of lactation, making it a naturally concentrated milk [3]. In comparison with milk from other dairy cows, yak milk possesses unique bioactive compounds, such as immunoglobulins and lactoferrin, which have resulted from its prolonged adaptation to cold and hypoxic environments [4]. Studies have shown that these substances can enhance human immunity, prevent diseases, and improve skin quality [5]. With the popularity of a healthy diet and the continuous upgrading of consumer demand, consumer demand for dairy products has gradually changed from “increasing milk intake” to “high-quality milk consumption”. Therefore, improving the yield and quality of yak milk is an urgent need and a major challenge facing the dairy industry.

Yak milk and its derived dairy products are becoming more and more popular because of their special nutritional value. It is a complex and arduous task to improve milk yield and optimize other characteristics of milk production, like fat and protein content. This process is affected by factors such as yak breed characteristics, feeding management, age, seasonal variation, and professional milking techniques [6]. These milk production traits are not only directly related to the economic value of yaks, but also become important indicators to evaluate their breeding effectiveness [7]. However, in the face of these complex genetic and environmental factors, traditional breeding methods have the limitations of being time-consuming and having low efficiency, making it hard to achieve the desired results efficiently. In recent years, with the development of molecular biology, a variety of molecular breeding methods have emerged, such as whole genome selection [8], high-throughput sequencing technology [9], chip technology [10] and molecular marker-assisted selection. Among them, single nucleotide polymorphism (SNP) markers, as the third-generation molecular marker technology of molecular marker-assisted selection, have become an indispensable tool in the field of molecular marker-assisted breeding due to their large number, wide distribution, and high genetic stability [11]. *CRTAC1*, a glycosylated extracellular matrix protein, has received much attention in animal genetics and dairy science in recent years. Bart Buitenhuis et al. [12] found that *the CRTAC1* gene was significantly associated with the fatty acid content of the milk of Danish Holstein and Jersey cattle, suggesting that it may be a key candidate gene for improving milk quality. Further results showed that the *CRTAC1* protein not only appeared in colostrum but also continued to be expressed in the mature milk stage of lactation [13]. The gene is widely distributed in whey protein throughout the lactation stage [14]. Other studies have shown that mutations in *CRTAC1* are associated with hind udder height traits in Holstein cattle [15,16], and RNA sequencing results have shown that this gene is related to mammary gland development in dairy cows [17]. However, although the *CRTAC1* gene has made some important advances in the milk traits of animals, the specific mechanism of action in lactation still needs to be further studied in detail.

The *IQGAP2* gene belongs to the gtpase activating protein (GAP) family and is involved in cell signaling pathways. In the yak Bosgru v3.0 reference genome (GCA_005887515.1), the *IQGAP2* gene is located on chromosome 11. Zhou et al. found that chromosome 11 has a significant effect on milk protein composition in Holstein cows [18]. In addition, the gene indirectly affects beef tenderness [19]. In Wang and Deng’s study, *IQGAP2* is involved in regulating abdominal fat content and disease resistance in chickens [20,21]. Additional research has demonstrated that the expression level of the *IQGAP2* gene influences the incidence of breast cancer [22]. *IQGAP2* acts as a tumor suppressor in breast cancer [23], limiting the malignant proliferation and invasive capacity of breast cancer cells by inhibiting the IQGAP1-mediated activation of ERK (extracellular signal-regulated kinase), further attenuating the oncogenic properties of the cells by regulating the MEK-ERK and p38 signaling pathways, thus maintaining the health of breast tissue [24,25,26]. The in-depth investigation of the *IQGAP2* gene offers novel insights for enhancing production efficiency and refining product quality within the dairy industry.

The Gannan yak, a breed indigenous to the Gannan region of Gansu Province in Northwest China, thrives at an average altitude exceeding 2800 m and serves as a valuable milk source for local herdsmen. Compared with Holstein and Jersey cows, Gannan yak milk is favored for its high protein, high fat, high energy, and rich vitamin and mineral content [1,27,28]. However, due to their remote geographical location, the collection of yak milk is difficult. The manual milking method is inefficient, and yak milk production is relatively low. These problems remain unsolved, resulting in its potential value not being fully explored and utilized [29]. Currently, there is a lack of research examining the relationship between the *IQGAP2* and *CRTAC1* genes and milk quality traits in Gannan yaks. Consequently, the present study was designed to investigate the polymorphic variations within the *IQGAP2* and *CRTAC1* genes and to explore their associations with milk quality in Gannan yaks. The ultimate goal of this research is to provide a scientific and theoretical foundation for enhancing the milk quality of this valuable breed.

## 2. Materials and Methods

### 2.1. Ethics Approval

The animal experiments were all authorized by the Lanzhou Institute of Animal Husbandry and Pharmaceutical Sciences, Chinese Academy of Agricultural Sciences (CAAS), under grant number 1610322020018.

### 2.2. Experimental Animals and Milk Composition Analysis

The test sample was the milk of the Gannan yak (Xiahe County, Gannan Tibetan Autonomous Prefecture. Located at 34.99° N, 102.92° E). The calving parity of the yaks ranged from 2 to 3. A total of 162 milk samples were collected for the purpose of milk composition analysis. A comprehensive analysis of the composition of yak milk was conducted utilizing the MilkoScan™ FT120 milk analyzer, manufactured by FUCHS Analytical Instruments Ltd. in Hellerup, Denmark. The analysis encompassed a range of indicators, including casein, protein, fat content, non-fat milk solids (SNF), lactose, and total solids (TS).

### 2.3. DNA Sample Extraction

From 162 Gannan yaks, ear tissue samples were collected and preserved in liquid nitrogen. Utilizing the magnetic bead method (DP341 model, supplied by Tiangen Biochemical Technology Co., Ltd., Beijing, China) and adhering to the specific instructions provided with the kit, DNA was extracted from these ear tissues. The concentration of the extracted DNA was measured using a Qubit fluorometric quantitation instrument 4.0 (Thermo Field, Nashville, TN, USA), while the integrity of the DNA samples was assessed via 2% gel electrophoresis.

### 2.4. Genotyping 

A total of 162 Gannan yaks underwent a comprehensive genotyping analysis using the Illumina Yak cGPS 7K chip (Illumina, Huazhi Biotechnology Co., Ltd., Changsha, China). This chip is designed to employ synthetic specificity probes capable of capturing and enriching multiple distinct target sequences at unused genomic loci through liquid-phase hybridization techniques. Following this enrichment process, second-generation sequencing was implemented to sequence the captured target intervals, thereby obtaining the genotypes of all SNP and insertion/deletion (INDEL) sites within these intervals. To ensure the quality of the raw sequencing data, Fastp was employed. Reads were filtered out if they contained a proportion of bases with a quality score of Q ≤ 20 exceeding 50% of the total bases, resulting in the exclusion of the entire read pair. Furthermore, reads harboring an excessive number of 5 N bases or with a length of less than 100 bases were also excluded from the analysis. The genomic positions of the SNPs were determined by aligning them to the yak reference genome assembly, Bosgru v3.0 (GCA_005887515.1).

### 2.5. Data Statistics and Analysis

Genetic heterozygosity (He), the effective number of alleles (Ne), polymorphism information content (PIC), and genotype and allele frequencies were calculated using GDICALL online software (http://www.msrcall.com/Gdicall.aspx) (last accessed on 9 August 2024). The *p* values were obtained, and the Hardy–Weinberg equilibrium (HWE) test was performed. The general linear model (GLM) in IBM SPSS Statistics 25 (IBM, Armonk, NY, USA) was used to evaluate the correlation between polymorphisms in the *IQGAP2* and *CRTAC1* genes and the milk quality traits of Gannan yaks. The simplified model is represented by Equation (1), where Yi denotes the phenotypic value of a milk quality trait, µ represents the population mean for that trait, SNPi signifies the fixed effects of different genotypes at specific loci, and e represents the random error. Multiple comparisons were conducted using the least significant difference (LSD) method, and the results are presented as the mean ± standard deviation.
Yi = µ + SNPi + e,(1)

## 3. Results

### 3.1. IQGAP2 and CRTAC1 Genotyping Results and Genetic Parameter Analysis

The genotyping results of the *IQGAP2* and *CRTAC1* genes in Gannan yaks, conducted using the Illumina Yak cGPS 7K liquid chip, revealed that the *IQGAP2* gene is located on chromosome 11 of the yak reference genome Bosgru v3.0 (GCA_005887515.1), with mutation sites of g.232,769C>G and g.232,922G>C. The *CRTAC1* gene, located on chromosome 25, harbors mutation sites of g.4,203T>C, g.5,348T>G, and g.122,451T>C. The genetic parameters of these SNP loci, as presented in Table 1, indicate that there are three genotypes for each of the five SNP loci across 162 Gannan yak samples. At the g.232769C>G mutation site, the allele frequencies of C and G were 0.685 and 0.315, respectively, with C being the dominant allele. At the g.232,922G>C mutation site, the frequencies of the GG, GC, and CC genotypes were 0.501, 0.382, and 0.117, respectively, indicating that the GG genotype was the most prevalent among these three. At the mutation sites of the *CRTAC1* gene, g.4,203T>C and g.122,451T>C, the frequency of allele C is higher than that of T, indicating that C is the dominant allele at these sites. At the g.5,348T>G locus of the *CRTAC1* gene, the frequency of the TT genotype is higher than that of both the TG and GG genotypes, suggesting that the TT genotype is predominant at this locus. The five loci (g.232,769C>G, g.232,922G>C, g.4,203T>C, g.5,348T>G, and g.122,451T>C) are moderately polymorphic (0.25 < PIC < 0.5) and are in Hardy–Weinberg equilibrium (*p* > 0.05).

### 3.2. Association Analysis of IQGAP2 and CRTAC1 Gene Polymorphisms with Milk Quality Traits in Gannan Yaks

The association analysis results between the SNPs in the *IQGAP2* and *CRTAC1* genes and the milk quality traits of Gannan yaks are presented in Table 2. The SNP locus g.232,769C>G within the *IQGAP2* gene exhibits a significant correlation with lactose traits (*p* < 0.05). Individuals possessing the GC genotype display a significantly elevated lactose content compared to those with the GG genotype. No statistically significant differences were observed between the other milk quality traits and the three genotypes at this locus (*p* > 0.05). The g.232,922G>C SNP site demonstrates a significant association with casein, protein, fat, non-fat milk solids, lactose, and total solids (*p* < 0.05). Individuals with the GG genotype exhibit significantly higher levels of casein and protein compared to those with the CC genotype. Individuals with the CC genotype show significantly higher fat and total solid contents compared to those with the GC genotype. Furthermore, individuals with either the GG or GC genotypes possess significantly greater non-fat milk solids content than those with the CC genotype. Notably, individuals carrying the GC genotype also exhibit a significantly higher lactose content than those with the CC genotype.

*CRTAC1* gene’s g.4,203T>C locus exhibited a significant correlation with protein, fat, and non-fat milk solids traits (*p* < 0.05). Yaks with the TT genotype demonstrated a significantly higher protein content than those with the TC genotype. Yaks possessing the CC genotype had a significantly elevated fat content compared to those with the TT genotype. Yaks with the TT genotype showed a significantly higher non-fat milk solids content than those with the CC genotype. The g.5,348T>G locus displayed a significant association with casein, protein, non-fat milk solids, and lactose traits (*p* < 0.05). Yaks with the TT genotype exhibited a significantly higher casein content compared to those with the GG genotype. Individuals with the TT genotype had a significantly higher protein and non-fat milk solids content than both TG and GG genotype yaks. Yaks with the TG genotype showed a significantly higher lactose content than those with the TT genotype. The g.122,451T>C locus was significantly associated with protein, fat, and non-fat milk solids traits (*p* < 0.05). Yaks with the TT genotype demonstrated a significantly higher protein and non-fat milk solids content compared to those with the CC genotype. Yaks possessing the CC genotype had a significantly higher fat content than both TT and TC genotype yaks. No statistically significant differences were observed for other phenotypic traits among the three genotypes at this particular locus (*p* > 0.05).

## 4. Discussion

Milk quality traits are the most important economic characteristics in dairy farming, and their optimization and improvement are crucial for the industry’s development. Yak milk quality traits are regulated jointly by multiple genes with micro-effects. This provides a broad pathway for the application of molecular breeding technology. SNPs, as an important tool in molecular breeding, offer favorable support for precisely improving milk quality traits. In this study, five SNP loci were identified in the *IQGAP2* and *CRTAC1* genes of Gannan yaks, including g.232,769C>G and g.232,922G>C of the *IQGAP2* gene, as well as g.5,348T>G g.4,203T>C and g.122,451T>C of the *CRTAC1* gene. After genetic polymorphism analysis of these five SNP loci, it was found that their PIC values were between 0.25 and 0.5, indicating moderate polymorphism. The PIC value, which serves as an important indicator of the genetic variation at SNP loci, directly reflects the degree of genetic variation and selection potential of a population [30], and is positively correlated with the selection space and improvement potential. This indicates that these 5 SNP loci have corresponding genetic variation stability and population representativeness in the Gannan yak population and show a certain selection potential. In addition, these SNP loci were in Hardy–Weinberg equilibrium, further indicating that their genetic structure in the Gannan yak population was relatively stable.

Yak milk is an important dairy resource in China, and its lactose content (4.5% to 5.0%) is higher than that of ordinary milk (about 3.4%) and closer to human milk (about 7.4%) [31]. Wu et al.’s [32] research elucidated a pivotal role for the *IQGAP2* gene in inhibiting the epithelial–mesenchymal transition (EMT) process in breast cancer through modulation of the Wnt/β-catenin signaling pathway [22], reinforcing its status as a breast cancer suppressor [23]. Our current study delves deeper into the correlation between the *IQGAP2* gene and milk quality in Gannan yaks. We observed a significant association between two specific loci within the *IQGAP2* gene (g.232,769C>G and g.232,922G>C) and the milk quality of Gannan yaks. Yaks harboring heterozygous or mutant genotypes at these loci exhibited enhanced contents of fat, lactose, and total solids in their milk, resulting in a marked improvement in the overall milk quality. This discovery underscores the fact that the *IQGAP2* gene not only impacts the health of the mammary gland but also indirectly modulates the quantity and quality of milk secretion by influencing breast development. Mutations in the *IQGAP2* loci significantly boosted the lactose, milk fat, and total solids content in Gannan yak milk, thereby enriching its nutritional profile. Lactose, a vital component of milk, fulfills various physiological roles. In the small intestine, it is hydrolyzed by lactase into glucose and galactose, serving as an energy source for the body [1]. Furthermore, it acts as a carbon source for lactic acid bacteria fermentation, aiding in maintaining the gut’s acidic environment and promoting microbial flora balance [33]. For newborns, lactose is indispensable for fostering a healthy gut microbiome, which is crucial for growth, development, and bone health [34,35]. In summary, the mutation in the *IQGAP2* gene and its associated loci hold significant promise for enhancing milk quality in Gannan yaks.

The present study further revealed significant associations between SNP loci of the *CRTAC1* gene (g.4,203T>C, g.5,348T>G, and g.122,451T>C) and the milk quality of Gannan yaks. Among them, the CC genotype of g.4,203T>C and the CC mutant genotype of g.122,451T>C had a positive effect on milk fat content, suggesting that the *CRTAC1* gene plays an important role in the synthesis or accumulation of milk fat. In addition, the TG genotype of g.5,348T>G significantly increased the lactose content. In studies by others, *CRTAC1* has been confirmed as a key candidate gene for improving the quality of Holstein and Jersey milk, which is closely related to the fatty acid content of milk [12]. This is consistent with the results of the present study, which further confirmed that the *CRTAC1* gene is associated with animal milk quality and also verified the importance of the *CRTAC1* gene in yak. Milk fat, as an important indicator to evaluate milk quality, is not only rich in fat-soluble vitamins but also plays an important role in promoting lipid metabolism and maintaining the health of the digestive system [36]. In yaks, milk fat mainly exists in the form of milk fat globals, which play a crucial role in promoting lipid metabolism and maintaining the health of the digestive system [37]. Due to its high milk fat content, this makes yak milk an ideal raw material for the production of quality butter, cheese, and other dairy products, which are not only an important source of nutrients in the daily diet but also an important way to provide energy and absorb fat-soluble vitamins for pastoral residents [38]. In addition, yak milk fat has a different fatty acid composition due to the unique grazing environment and is rich in beneficial fatty acids, including monounsaturated fatty acids (MUFA), polyunsaturated fatty acids (PUFA), and conjugated linoleic acids (CLA), which have positive effects on cancer and diabetes prevention and the health of organs such as the brain, heart, and eye [39]. Therefore, the above SNPs in the *CRTAC1* gene are of great significance for improving the milk quality of Gannan yaks.

We found that the *IQGAP2* (g.232,769C>G and g.232,922G>C) and *CRTAC1* (g.4,203T>C, g.5,348T>G, and g.122,451T>C) sites were located in the intron regions. SNPs in the intron regions affect the splicing and expression of genes, and may also serve as genetic markers to assist in the selection of individuals with superior traits, thereby accelerating the process of genetic improvement [40]. Li et al. [41] discovered a significant correlation between the SNP site located in intron 2 of the *GHR* gene and the concentration of C18:0 in milk. Fifteen SNPs located within the intron of the buffalo *STAT1* gene were found to be associated with both protein percentage and milk yield [42]. This is consistent with the results of this study. This indicates that the five SNP loci identified in this study can effectively affect the milk quality traits of the Gannan yak.

With advancements in molecular biology, SNP loci have emerged as selective markers for enhancing product quality [43]. Research has demonstrated that variations in the *FASN* genes of yaks are correlated with the percentages of milk fat and total milk solids [44]. Additionally, newly identified polymorphisms within the *SLC2A12* and *SLC5A1* genes have been found to influence milk yield, as well as the protein, fat, and lactose contents in Holstein Friesian cows [45]. In our study, significant associations were found between the SNP loci of the *IQGAP2* and *CRTAC1* genes and milk quality traits in Gannan yaks. It was found that heterozygous and mutant populations at the g.232,769C>G, g.232,922G>C, g.5,348T>G, g.4,203T>C, and g.122,451T>C loci significantly increased the lactose, TS, and milk fat contents of Gannan yaks. Therefore, the above five SNP loci can be used as molecular marker loci for improving milk quality in Gannan yaks. Screening individuals carrying specific SNP loci has proven to be an effective approach for developing yak varieties with superior traits [46]. These SNP loci allow for the preferential identification of yaks that carry favorable alleles, greatly facilitating the screening process and significantly accelerating the breeding progress for improvement.

## 5. Conclusions

In this study, we identified five SNP loci within the *IQGAP2* (g.232,769C>G and g.232,922G>C) and *CRTAC1* (g.4,203T>C, g.5,348T>G, and g.122,451T>C) genes of Gannan yaks. An association analysis demonstrated that these SNP loci had a positive impact on milk fat, lactose, and total solids (TS) content in Gannan yak milk. These five SNPs can serve as molecular markers for improving the milk quality of Gannan yaks.

## Figures and Tables

**Table 1 foods-13-03720-t001:** The population genetic parameters of SNPs loci in *IQGAP2* and *CRTAC1* genes of Gannan yaks.

SNPs	Genotypic Frequencies			Allelic Frequencies		He	Ne	PIC	*p*-Value
g.232,769C>G	CC (79)	CG (64)	GG (19)	C	G	0.395	0.605	0.431	1.15
	0.487	0.395	0.118	0.685	0.315				
g.232,922G>C	GG (81)	GC (62)	CC (19)	G	C	0.383	0.617	0.426	1.726
	0.501	0.382	0.117	0.691	0.309				
g.4,203T>C	TT (18)	TC (69)	CC (75)	T	C	0.426	0.574	0.438	0.125
	0.111	0.426	0.463	0.324	0.676				
g.5,348T>G	TT (95)	TG (57)	GG (10)	T	G	0.352	0.648	0.362	0.136
	0.586	0.352	0.062	0.762	0.238				
g.122,451T>C	TT (24)	TC (79)	CC (59)	T	C	0.488	0.512	0.477	0.769
	0.148	0.488	0.365	0.392	0.608				

Note: He—heterozygosity; Ne—effective number of alleles; PIC—polymorphism. PIC < 0.25, low polymorphism; 0.25 < PIC < 0.5, moderate polymorphism; PIC > 0.5, high polymorphism; *p* > 0.05 suggests that the population gene is in Hardy–Weinberg balance, and the sample comes from the same Mendel population.

**Table 2 foods-13-03720-t002:** Association analysis of *IQGAP2* and *CRTAC1* gene polymorphisms with milk quality traits in Gannan yaks.

SNPs g. 232,769C>G						
**Genotype**	**Casein/%**	**Protein/%**	**Fat/%**	**SNF/%**	**Lactose/%**	**TS/%**
CC 79	4.14 ± 0.29	4.96 ± 0.39	5.70 ± 3.06	11.32 ± 0.5	4.97 ± 0.16 ab	16.9 ± 2.99
GC 64	4.05 ± 0.28	4.86 ± 0.39	5.14 ± 2.20	11.27 ± 0.44	5.02 ± 0.16 a	16.28 ± 2.15
GG 19	4.01 ± 0.31	4.79 ± 0.44	6.39 ± 1.89	11.02 ± 0.45	4.91 ± 0.13 b	17.35 ± 1.92
*p* Value	*p* = 0.076	*p* = 0.135	*p* = 0.157	*p* = 0.053	*p* = 0.017	*p* = 0.184
SNPs g. 232,922G>C						
**Genotype**	**Casein/%**	**Protein/%**	**Fat/%**	**SNF/%**	**Lactose/%**	**TS/%**
GG 81	4.15 ± 0.03 a	4.97 ± 0.04 a	5.69 ± 0.29 ab	11.33 ± 0.05 a	4.97 ± 0.02 ab	16.89 ± 0.29 ab
GC 62	4.04 ± 0.04 ab	4.84 ± 0.05 ab	5.15 ± 0.33 b	11.26 ± 0.06 a	5.02 ± 0.02 a	16.28 ± 0.33 b
CC 19	3.96 ± 0.072 b	4.69 ± 0.1 b	6.78 ± 0.66 a	10.91 ± 0.12 b	4.93 ± 0.04 b	17.66 ± 0.65 a
*p* Value	*p* = 0.017	*p* = 0.018	*p* = 0.019	*p* = 0.005	*p* = 0.023	*p* = 0.014
SNPs g.4,203T>C						
**Genotype**	**Casein/%**	**Protein/%**	**Fat/%**	**SNF/%**	**Lactose/%**	**TS/%**
TT 18	4.19 ± 0.24	5.08 ± 0.33 a	4.6 ± 2.02 b	11.56 ± 0.35 a	5.02 ± 0.16	15.99 ± 1.97
TC 69	4.03 ± 0.3	4.82 ± 0.41 b	5.58 ± 2.6 ab	11.19 ± 0.48 b	4.99 ± 0.16	16.46 ± 2.53
CC 75	4.12 ± 0.29	4.93 ± 0.38 ab	6.00 ± 2.73 a	11.27 ± 0.47 b	4.97 ± 0.15	17.1 ± 2.73
*p* Value	*p* = 0.056	*p* = 0.029	*p* = 0.043	*p* = 0.011	*p* = 0.425	*p* = 0.155
SNPs g. 5,348T>G						
**Genotype**	**Casein/%**	**Protein/%**	**Fat/%**	**SNF/%**	**Lactose/%**	**TS/%**
TT 95	4.28 ± 0.09 a	5.23 ± 0.33 a	4.69 ± 1.89	11.64 ± 0.36 a	4.92 ± 0.12 b	16.14 ± 1.9
TG 57	4.01 ± 0.04 ab	4.77 ± 0.41 b	5.66 ± 2.64	11.18 ± 0.52 b	5.02 ± 0.15 a	16.5 ± 2.56
GG 10	4.12 ± 0.03 b	4.94 ± 0.37 b	5.77 ± 2.68	11.27 ± 0.44 b	4.97 ± 0.16 ab	16.89 ± 2.67
*p* Value	*p* = 0.009	*p* = 0.001	*p* = 0.465	*p* = 0.019	*p* = 0.041	*p* = 0.518
SNPs g.122,451T>C						
**Genotype**	**Casein/%**	**Protein/%**	**Fat/%**	**SNF/%**	**Lactose/%**	**TS/%**
TT 24	4.16 ± 0.24	5.07 ± 0.34 a	4.84 ± 2.01 b	11.46 ± 0.38 a	4.94 ± 0.14	16.13 ± 1.96
TC 79	4.08 ± 0.3	4.91 ± 0.41 ab	5.35 ± 2.35 b	11.3 ± 0.47 ab	4.99 ± 0.16	16.52 ± 2.36
CC 59	4.07 ± 0.3	4.82 ± 0.38 b	6.48 ± 3.04 a	11.15 ± 0.5 b	5 ± 0.16	17.19 ± 3.02
*p* Value	*p* = 0.451	*p* = 0.029	*p* = 0.019	*p* = 0.012	*p* = 0.329	*p* = 0.159

Note: In the same column of data, different shoulder letters indicated significant difference (*p* < 0.05), and the data were expressed as “mean ± standard deviation”. IQGAP2: g.232,769C>G, g.232,922G>C. CRTAC1: g.4,203 T>C, g.5,348 T>G, g.122,451T>C.

## Data Availability

The original contributions presented in the study are included in the article; further inquiries can be directed to the corresponding author.

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
