# Peer review of "Polymorphisms Within the IQGAP2 and CRTAC1 Genes of Gannan Yaks and Their Association with Milk Quality Characteristics"

_foods, 2024, doi:10.3390/foods13233720_

Round 1

Reviewer 1 Report

Comments and Suggestions for Authors

Manuscript: foods-3295626

Title: Polymorphisms within the IQGAP2 and CRTAC1 Gene of Gannan Yaks and Their Association with Milk Quality Characteristics

In this study, the authors genotyped the Yaks' DNA to find SNPs related to milk traits. Specifically, they analyzed the IQCAP2 and CRTAC1 genes, finding five SNPs correlated with lactose and total fat. Considering that Yak animals are essential in the economy of people from Tibet and because little information exists about them, the results presented in this document are necessary to improve the breeding protocols of these animals.

In the Introduction section, I have just one comment:

Lines 72 – 73: The authors describe 9 SNP in ITGA9, but it is unclear what this influence is in the study. What is the relation between this gene and the objective of the study? The authors must clarify this idea or delete it.

Author Response

Comments 1:Lines 72 – 73: The authors describe 9 SNP in ITGA9, but it is unclear what this influence is in the study. What is the relation between this gene and the objective of the study? The authors must clarify this idea or delete it.

Response 1:Based on your suggestions ,we have decided to remove that sentence from the introductory section to ensure a more polished and fluid text (Lines 72-73).

Reviewer 2 Report

Comments and Suggestions for Authors

This manuscript investigates the association between single nucleotide polymorphisms (SNPs) in the IQGAP2 and CRTAC1 genes and milk quality traits in Gannan yaks. With the increasing interest in molecular breeding of livestock, this study provides relevant insights into potential genetic markers that could improve milk quality in Gannan yaks, with particular emphasis on milk fat, lactose and total solids (TS) content. The study makes a significant contribution to our understanding of the genetic factors influencing milk quality and highlights key SNPs as potential markers for future breeding programmes.

The use of the Illumina Yak cGPS 7K liquid chip to genotype 162 yaks provides a solid methodological basis. The authors carefully identified and analysed five SNPs in the IQGAP2 and CRTAC1 genes, ensuring the reliability of the data and subsequent analysis. The study effectively combines polymorphism analysis with Hardy-Weinberg equilibrium testing and association analysis to confirm that the identified SNPs are relevant and moderately polymorphic. This approach strengthens the validity of the conclusions by showing a robust association between these polymorphisms and milk quality traits.

The results clearly show that specific genotypes at the loci g.232,769C>G and g.232,922G>C in the IQGAP2 gene, and g.4,203T>C, g.5,348T>G and g.122,451T>C in the CRTAC1 gene are associated with higher milk fat, lactose and TS content. This association provides actionable data that could support selective breeding efforts to improve milk quality in Gannan yaks. By identifying SNPs with a positive effect on milk quality, the authors provide a valuable basis for molecular breeding. The results could be practically applied to improve milk production traits, which is particularly useful for the dairy industry focused on yak milk products.

I have made some comments about things I think you could do to improve your work. This does not mean that you have to agree or rewrite in the same way. It is just a suggestion and a different view, with the aim of contributing.

1) While the study presents a robust association between SNPs and milk quality traits, it would benefit from a discussion of the potential mechanisms by which these genes influence milk composition. In particular, the role of IQGAP2 in tumour suppression and CRTAC1 in fatty acid correlation suggest underlying pathways that may be relevant to milk quality, which could be discuss to provide additional context;

2) A brief comparison with similar studies on other yak or cattle breeds would strengthen the manuscript. By contextualising these findings within the wider field, readers could gain a better understanding of the unique or common characteristics of Gannan yaks;

3) The authors could consider extending the discussion on how these results could be integrated into breeding programmes. Specific recommendations on the use of these SNP markers in selective breeding for desired milk traits would enhance the practical utility of the study;

4) Although the manuscript reports significant associations with P values < 0.05, a more detailed explanation of statistical adjustments (if any) for multiple comparisons could help clarify the robustness of these associations. This would ensure the validity of conclusions drawn from multiple SNP analyses.

The manuscript is well written with a clear and concise presentation of the results. The study is well designed, with statistically significant results that have clear implications.

The paper could be considered as it stands or with minor modifications (if the authors so wish).

Author Response

Comments 1: While the study presents a robust association between SNPs and milk quality traits, it would benefit from a discussion of the potential mechanisms by which these genes influence milk composition. In particular, the role of IQGAP2in tumour suppression and CRTAC1 in fatty acid correlation suggest underlying pathways that may be relevant to milk quality, which could be discuss to provide additional context;

Response 1: Thank you very much for your careful review of our study and your valuable suggestions. We have supplemented the potential mechanisms by which the IQGAP2 and CRTAC1 genes affect milk quality in the introduction section (lines 73-84 and 93-96).

Comments 2: A brief comparison with similar studies on other yak or cattle breeds would strengthen the manuscript. By contextualising these findings within the wider field, readers could gain a better understanding of the unique or common characteristics of Gannan yaks;

Response 2: Thank you very much for your comments. We have revised the manuscript by adding a comparative study of Gannan yak with other cattle breeds (e.g. Holstein and Juanzan) in terms of milk quality in the introduction section. Through this comparison, we aim to further highlight the unique advantages and qualities of Gannan yak milk (Line 102-104).

Comments 3: The authors could consider extending the discussion on how these results could be integrated into breeding programmes. Specific recommendations on the use of these SNP markers in selective breeding for desired milk traits would enhance the practical utility of the study;

Response 3: Thank you for your proposal, and in the revised version, we have added this section in the discussion section, where we emphasize the importance of marker-assisted selection using the SNP markers identified in this study. With genotyping techniques, breeders can identify individuals carrying superior alleles and thus perform selective breeding more efficiently and accelerate the genetic progress of milk quality traits (Line312-317).

Comments 4: Although the manuscript reports significant associations with P values < 0.05, a more detailed explanation of statistical adjustments (if any) for multiple comparisons could help clarify the robustness of these associations. This would ensure the validity of conclusions drawn from multiple SNP analyses.

Response 4: In this study, we prudently chose the general linear model (GLM) as a statistical tool to explore the potential relationship between genotype and milk quality traits, a decision based on a comprehensive review of the existing literature and the published results of our group's research in the field. The advantage of the GLM as a well-recognised statistical tool lies in its ability to accurately analyse the linear correlation between the independent variable (genotype) and the dependent variable (milk quality traits), and to properly incorporate other potential covariates within the analytical framework to ensure the accuracy and comprehensiveness of the results. Accordingly, we are confident that the findings of this study are statistically valid.
[1] Yang, G.; Zhang, J.; Ma, X.; Ma, R.; Shen, J.; Liu, M.; Yu, D.; Feng, F.; Huang, C.; Ma, X.; et al. Polymorphisms of CCSER1 Gene and Their Correlation with Milk Quality Traits in Gannan Yak (Bos grunniens). Foods, 2023, 12, 4318. https://doi.org/10.3390/foods12234318
[2] Ma, X.; Yang, G.; Zhang, J.; Ma, R.; Shen, J.; Feng, F.; Yu, D.; Huang, C.; Ma, X.; La, Y.; et al. Association between Single Nucleotide Polymorphisms of PRKD1 and KCNQ3 Gene and Milk Quality Traits in Gannan Yak (Bos grunniens). Foods 2024, 13, 781. https://doi.org/10.3390/foods13050781

[3] Feng, F.; Yang, G.; Ma, X.; Zhang, J.; Huang, C.; Ma, X.; La, Y.; Yan, P.; Zhandui, P.; Liang, C. Polymorphisms within the PRKG1 Gene of Gannan Yaks and Their Association with Milk Quality Characteristics. Foods 2024, 13, 1913. https://doi.org/10.3390/foods13121913

Reviewer 3 Report

Comments and Suggestions for Authors

The manuscript presents the results of a study in which certain genotypes that may be related to the quality of Jack's milk are being searched for. The introduction highlights the importance of the milk of this species in the regions where it is raised. The methodology is well laid out and the results are clear. On the other hand, the discussion needs to be rewritten; the authors practically limit themselves to re-presenting the results and hardly carry out a comparative analysis with what is known in other species. This should be corrected so that the work can be considered for the Journal.

Author Response

Comments 1:The manuscript presents the results of a study in which certain genotypes that may be related to the quality of Jack's milk are being searched for. The introduction highlights the importance of the milk of this species in the regions where it is raised. The methodology is well laid out and the results are clear. On the other hand, the discussion needs to be rewritten; the authors practically limit themselves to re-presenting the results and hardly carry out a comparative analysis with what is known in other species. This should be corrected so that the work can be considered for the Journal.

Response 1: Thank you very much for your thorough review and recognition of our work, especially for the commendation on the clarity of our research methodology and the precise presentation of our results. In response to your feedback, we have rewritten the discussion section. We have incorporated similar research findings to ours and discussed the commonalities with our study. Additionally, beyond merely presenting the results, we have conducted a deeper analysis and interpretation of our findings, including possible underlying mechanisms and potential applications (Line 247-317).

Round 2

Reviewer 3 Report

Comments and Suggestions for Authors

In the present form the manuscript can be accept-